# Evaluation of Full-Body Gestures Performed by Individuals with Down Syndrome: Proposal for Designing User Interfaces for All Based on Kinect Sensor

**DOI:** 10.3390/s20143930

**Published:** 2020-07-15

**Authors:** Marta Sylvia Del Rio Guerra, Jorge Martin-Gutierrez

**Affiliations:** 1Department of Computer Science, Universidad de Monterrey, Nuevo León 66238, Mexico; marta.delrio@udem.edu; 2Department of Techniques and Projects in Engineering and Architecture, Universidad de La Laguna, 38071 Tenerife, Spain

**Keywords:** Microsoft Kinect, sensor, corporal gestures, full-body gestures, user interface, user experience, evaluation, Down syndrome, human–computer interaction

## Abstract

The ever-growing and widespread use of touch, face, full-body, and 3D mid-air gesture recognition sensors in domestic and industrial settings is serving to highlight whether interactive gestures are sufficiently inclusive, and whether or not they can be executed by all users. The purpose of this study was to analyze full-body gestures from the point of view of user experience using the Microsoft Kinect sensor, to identify which gestures are easy for individuals living with Down syndrome. With this information, app developers can satisfy Design for All (DfA) requirements by selecting suitable gestures from existing lists of gesture sets. A set of twenty full-body gestures were analyzed in this study; to do so, the research team developed an application to measure the success/failure rates and execution times of each gesture. The results show that the failure rate for gesture execution is greater than the success rate, and that there is no difference between male and female participants in terms of execution times or the successful execution of gestures. Through this study, we conclude that, in general, people living with Down syndrome are not able to perform certain full-body gestures correctly. This is a direct consequence of limitations resulting from characteristic physical and motor impairments. As a consequence, the Microsoft Kinect sensor cannot identify the gestures. It is important to remember this fact when developing gesture-based on Human Computer Interaction (HCI) applications that use the Kinect sensor as an input device when the apps are going to be used by people who have such disabilities.

## 1. Introduction

Nowadays, users can interact with devices without needing to physically touch them. Applications use different sensors and artificial vision algorithms to interpret interaction gestures. Both industrial and research communities are showing an ever-growing interest in the practical applications of skeleton-based human action and full-body gesture recognition following the integration of depth sensors in human–machine interactions. Using gesture recognition software, these sensors allow users to communicate with machines using physical gestures. The reality is such sensors are becoming increasingly commonplace; they can now be found in the digital assistants used by many in their day-to-day lives, in human–robot interactions, or equally, in video-based monitoring and surveillance, amongst others [1].

By way of example, we interact with sensors such as the Microsoft Kinect to give instructions using full-body gestures when playing Active Video Games (AVG) [2], when visiting interactive museum exhibitions [3], or when using digital whiteboards in the classroom [4]. Such sensors are also incredibly useful in the field of healthcare; the Microsoft Kinect sensor can be used as part of a motion capture system to assist with physical exercise, or likewise with physical therapy and patient rehabilitation. Using the Microsoft Kinect sensor, researchers have proposed an architecture for a telerehabilitation platform. The platform in question offers precise physiotherapeutic information and musculoskeletal rehabilitation in real-time, thereby helping accelerate the recovery time of patients [5].

Continuing in the field of healthcare, numerous researchers have focused their attention on developing serious games using the Microsoft Kinect sensor to improve or cure certain illnesses or rehabilitate patients. All these researchers share one goal in common: to create a series of exergames (exercise-based videogames) based on AVG that deliver the necessary physical exercises required as part of a patient’s treatment plan [6,7,8]. Using the Microsoft Kinect sensor as an input device, Proffitt, Sevick, Chang, and Lange developed a customized controller-free videogame for hand rehabilitation [9]. Players were required to perform various gestures to accomplish a virtual cooking task to demonstrate the potential of the videogame as a method for delivering hand rehabilitation for a diverse population.

Pagliano et al. proposed a videogame-directed exercise program targeting children and adolescents diagnosed with Charcot–Marie–Tooth disease type 1A (CMT1A) [10]; using an Xbox 360 console connected to the Kinect movement sensor, the Kinect videogame delivers directed physical exercise activities aimed at improving overall balance and endurance by improving limb strength in the ankle.

A study of patients with schizophrenia suggests that a video game-based physical activity program provides a clinically meaningful improvement in their health status [11]. A study of cancer patients conducted by Oliveira et al. uses the Xbox 360 with the Microsoft Kinect and the commercially sold game ‘Your Shape Fitness Evolved’ (2012), to explore the effects of exergame protocol on patient disability scores and pain levels [12]. The findings revealed that the proposed exergaming protocol was effective in reducing reported symptoms and increasing the perceived quality of life.

A search of the scientific literature reveals the existence of a vast amount of research from different fields asserting the benefits of games that use Kinect sensors. For example, there is mention of the positive impact of AVG on children living with autism [13], and on the elderly when it comes to delivering fall prevention exercises [14].

In some cases, the research discussed has consisted of developing applications that require a Kinect sensor to either teach, or improve, or rehabilitate specific elements, i.e., sign language learning [15,16], cancer patient rehabilitation [12], Parkinson’s disease [17,18], localized therapeutic treatment [9], or overall health [6,19,20], to name a few. To ensure health and safety in the workplace and prevent occupational illness amongst workers on the factory floor resulting from poor posture, postural risk monitoring is performed using the application ErgoSentinel. In doing so, operators can correct poor posture and prevent injuries [21].

Furthermore, a significant amount of research has been conducted on the Microsoft Kinect sensor. This research permitted the creation of algorithms that unveiled to reveal the potential of this device [22,23,24]. As this sensor allows input instructions from both hardware and software, it is thus useful for object recognition and, equally, gesture recognition. In other words, the Microsoft Kinect sensor is an extremely versatile device, which makes it suitable for a wide range of applications [25,26]. Thus far, it has certainly has demonstrated its potential for Industry 4.0 [27,28].

Down syndrome (DS) is a genetic disorder caused by abnormal cell division resulting in an extra full copy or a partial copy of chromosome 21, which leads to both cognitive and physical impairments. Although individuals living with Down syndrome share common traits, on an individual level their own set of disabilities may range from mild to severe. Such differences can influence the success rate of gesture recognition by the Kinect sensor. The task of the sensor is to recognize a gesture (hand, corporal, facial) and send an instruction to the hardware or software. It should, therefore, be capable of recognizing a gesture regardless of the different morphological characteristics of a user, and despite variations in how a user may perform a gesture.

Devices such as the Microsoft Kinect sensor have been designed and developed with all users in mind. In many instances, however, the different applications supported by such devices have failed to make use of their potential for recognizing full-body gestures [25]. When employing these sensors, one should be mindful of which gestures are being elicited from users, and steps taken to make sure the gestures are suitable for all users [29].

Despite the widespread development of technologies that facilitate gestural interactions with IT programs using the hand, fingers, face, arms, legs, or body, as detailed in ISO 9241-960 [30], virtually no research has been conducted on the full-body gestures design from the perspective of User Interface Design (UI) perspective, and how well such gestures are recognized by the Microsoft Kinect sensor. Research is available, however, for other types of interactive gestures. Said research covers eye tracking, touch gestures, and also 3D mid-air gesture recognition [29,31].

The research described in this paper aims to identify which full-body gestures are identified by the Kinect sensor when they are performed by individuals with Down syndrome who demonstrate a range of physical and cognitive impairments. The findings will serve to establish which gestures are inclusive and suitable for all users, and this will provide software developers with a roadmap that can help them to select gestures during software development.

For this purpose, the research team developed an application to measure the battery of tests that were run with individuals living with DS of different ages (children, adolescents, and adults).

In this article, Section 2 provides details of related works that deal with gestures and DS. Section 3 presents the research objectives and hypotheses. Section 4 details the gesture selection process used by the research team. Section 5 contains a description of the materials and methodology used, together with details of the application that was designed and built to test and measure the selected gestures, the study population, and the experiment design. Section 6 contains the authors’ results and findings. Finally, conclusions and discussions are presented in Section 7. A guide is included in the annex section of this paper that provides an overall assessment of the twenty gestures that were the object of study.

## 2. Related Works

The gestural design has been the focal point of much research in recent years [32]. Gestures are being analyzed in greater depth now, as a result of the increase in the number of touch devices, wearables, urban interactive kiosks, touchscreens, and voice–user interfaces gestures are being analyzed more thoroughly to identify those that prove more intuitive, and thus more suitable for natural interface design. For this reason, the first question that should be addressed is ‘What makes a gesture more user-friendly or easier to execute?’ [33]. Wu and Kuo, for example, identified that touch gestures on a Smartphone are easier to perform if the gesture is performed from the top-down, rather than from the bottom-up (due to how the device is held). They also identified that horizontal finger movements are easier than vertical movements [34].

The study of pattern recognition in sensors such as the Kinect device has been widely addressed; so to have research studies involving individuals with cognitive and motor impairments in which the Kinect sensor plays a prominent role. The authors Wuang et al. studied the progress of sensorimotor skills of children using the Wii console and its controllers. However, they do not design or evaluate the gestures themselves, instead, they measure the results of the functions. In this case, their findings reveal that children in the group using the Wii show the greatest improvement in motor skills, and better visual–motor integration [35]. Other use examples can be found that include: a proposal of learning method via Microsoft Kinect to teach students with disabilities [36], to train children with intellectual disabilities on pedestrian safety issues [37]; to help individuals living with disabilities learn communication skills [38], or to train children with Down syndrome so they practice and improve visual-motor skills and cognitive abilities [39]. It is possible to continue listing different studies and research that involves individuals with Down syndrome or other cognitive or motor impairments and the use of Kinect sensor for different purposes, yet their focus is on the results of training/application and not on the evaluation of the full-body gestures themselves.

Some research is based on gesture recognition for specific applications, and specific algorithms have been developed for recognition. Verma, Aggarwal, and Chandra developed an algorithm that helps in translating sign language gestures performed by a user; to do this, the system must recognize the position of fingers, hands, and arms [40]. Cheng et al. created a system to recognize different body gestures and help non-expert users to control a Human-Robot, thereby making human–robot interaction much easier [41]. Emotion recognition was studied from body gestures using the Kinect sensor [42]. The proposed system identifies an emotion related to body gestures corresponding to five basic human emotional states: ‘Anger’, ‘Fear’, ‘Happiness’, ‘Sadness’, and ‘Relaxation’.

## 3. Objectives and Hypotheses

The objective of this research is to establish which of the full-body gestures are correctly executed by people with Down syndrome (individuals with impairments of the locomotor system and cognitive disabilities). The Kinect sensor is used to establish whether a gesture is executed correctly.

When designing applications that receive instructions through human interaction using gestures that can be recognized by the Kinect sensor, it is important to know which gestures can be correctly executed by people with disabilities to ensure the inclusive design is taken into consideration.

The underlying motivation for this research lays in the absence of user experience (UX) studies dealing with full-body gestures. In reviewing the scientific literature, no UX studies were found that deal with the validation of full-body gestures using individuals with DS on the grounds of their cognitive and motor impairments. As such, the goals of the research team are to (i) establish which gestures, from the set of full-body gestures selected for this study, are correctly executed by individuals with DS; (ii) establish which full-body gestures are the easiest for individuals with physical or cognitive impairments to execute.

Based on the analysis performed, the research team shall propose the gestures that are most suited to all users from a UX perspective. Consequently, it is these proposed gestures that should be considered by developers who are designing devices or tools based on full-body gesture recognition.

This work presents a study in which both the success rate and execution time of each gesture is analyzed. From these two sets of results, it is possible to establish which gestures are the most efficient. The study also examines whether the variable *Gender* affects success rates.

In this study, the function of the Kinect sensor is to indicate whether a user correctly executes a gesture. The user has 10 s to execute a gesture. If the Kinect sensor does not recognize the gesture within this time limit, the research team classifies the gesture as a failed gesture, that is to say, one that has been incorrectly executed by the user. If the Kinect sensor does recognize the gesture, it is classified as a successful gesture and the time taken to execute the gesture is logged in the database of the app developed for this study (see Section 5.1). The logged time is measured from the moment the user is shown the gesture that must be executed until the moment the Kinect recognizes the gesture as having been correctly executed.

Each gesture is executed as part of an action or task. The defined research hypotheses are as follows:

**H1.** 
*The presence or absence of Kinect recognition logs indicates that the different full-body gestures being executed by individuals with DS present different levels of execution complexity. Null Hypothesis H_01_: The different gestures executed by individuals with DS will present the same degree of difficulty.*


**H2.** 
*The execution success of full-body gestures is influenced by gender. It is believed that there is a significant difference between the rates of success and failure for gesture execution in terms of gender. Null Hypothesis H_02_: Gender does not influence the success of the execution of full-body gestures.*


**H3.** 
*Not all full-body gestures have the same success rate. In other words, the execution time is different, and some gestures will be recognized by the Kinect faster than others. Null Hypothesis H_03_: All full-body gestures have the same success rate.*


**H4.** 
*Gender influences the success rate of the execution of the different gestures. In other words, some gestures are easier to execute than others depending on the gender of the user. Null Hypothesis H_04_: Gender does not influence the success rate for different gestures.*


**H5.** 
*The time it takes for the user to execute the different full-body gestures is significant. In other words, some gestures are recognized faster than others. Null Hypothesis H_05_: All gestures take the same time to execute.*


**H6.** 
*Regarding success rates, the time taken to complete the different gestures differs for men and women. Men or women may execute gestures faster than the other gender. Null hypothesis: H_06_: Gender does not influence the execution time of full-body gestures.*


## 4. Creation and Selection of Gestures

Many applications allow users to interact with devices without needing to physically touch them. The Kinect sensor can recognize gestures and translate them into instructions within applications. Generally speaking, these gestures are developed using neurotypical individuals as a model. However, this gives rise to sensor errors in the form of recognition error or absence of recognition when such applications are used by individuals who experience some form of disability, whether it be in the form of motor or cognitive impairments.

By definition, full-body gestures involve the use of the entire body. The number of possible gestures to choose from is virtually unlimited; gestures can be defined for a single body segment such as the head, torso, or limbs, or equally any combination of these. For this study, the research team was able to narrow down the scope of full-body gestures to a list of twenty gestures. The selected gestures are those that are most commonly used to interact with any system that uses a movement recognition sensor as an input device. Table 1 below lists the selected gestures, their icons, and the corresponding visual descriptions.

### 4.1. Process Design

The process involved in designing a gesture for the Microsoft Kinect required the use of several tools [43]. Firstly, the research team used the Kinect Studio to record the clips containing each gesture, which are needed to obtain the Raw IR stream provided by the Kinect sensor. Importantly, this tool allowed each clip to be recorded separately. Having done so, the depth, IR, and body frame data streams could then be recorded from the sensor array. To generalize a gesture, a diverse variety of situations were included, e.g., different clothes, distances to the sensor, backgrounds, ways of performing the same gesture, and contexts where a certain gesture occurred. Furthermore, to train the system with negative examples, similar gestures that might be confused by the classifier were also recorded.

#### 4.1.1. Actors

The gestures were recorded using neurotypical actors. The research team has taken the stance that in any given application requiring gestures, neurotypical individuals are used as a model during the gesture development process. Nonetheless, anyone, even individuals with some degree of disability, should be able to perform these gestures. This study intends to identify effects on gesture recognition success when gestures are executed by individuals with DS.

Seven actors (4 male and 3 female) aged between 12–31 years old were used to record the gestures. Each actor was recorded performing each of the 20 gestures, with each gesture recorded 5 separate times. In other words, a total of 35 video clips were recorded for each gesture. Each participant, and actor, was recorded individually. This was to avoid individuals being influenced by others in terms of how gestures should be executed. The research team did not want participants to watch how other users perform the same gesture, or vice versa, and have this distract from onscreen instructions or influence behavior.

#### 4.1.2. Dataset Creation

Taking the raw data, Visual Gesture Builder (VGB) was used to build and train a total of 20 gestures. The accuracy of the machine-learning algorithms cannot be measured, but the VGB tool provides a data-driven solution for gesture detection through machine learning.

In this work, the Adaptive Boosting algorithm (AdaBoost) [44] was used to train all gestures. This AdaBoost algorithm has been widely used to study pose recognition as posture data [45,46,47] and it determines when the user/actor is performing a specific gesture. Such detection technology produces a binary or discrete result whether the actor is performing the gesture or not. During the training process, it accepted input tags—Boolean values—which marked the occurrence of a gesture. This marking or tagging is used to evaluate whether or not a gesture is actively happening and determines the confidence value of the event. Boosting is an approach to machine learning based on the idea of creating a highly accurate prediction rule by combining many relatively weak and inaccurate rules to build up a strong classifier.

For the training approach, the given input parameters for the algorithm are described below. The accuracy level value controlled how accurate the results were, but also affected the training time. The algorithm can potentially generate tens of thousands of weak classifiers. Using all of them increased accuracy [48]. It was necessary to apply a filter to the raw per frame results since the results of the algorithm are based on a frame, not on a gesture. The Adaptive Boosting algorithm that was used in VGB provides a simple low latency filter and is a simple sliding window of N frames, summing up the results and comparing them against a threshold value. The number of frames can be seen as a frequency and the threshold can be seen as amplitude.

The following settings were identical for each training gesture:Accuracy level: 0.95.Number of weak classifiers at run time: 0 (1000) to use all the classifiers generated.Filter result: True.Auto Find Best Filtering Params: True.Weight of False Positives During Auto Find: 0.5.Manual Filter Params Num Frames to Filter: 5.Manual Filter Params Threshold: 0.00.

Some of the gestures that were designed only need the algorithm to focus on arm movements, whilst ignoring lower-body segments. This reduced the amount of input data. In training, when using such an algorithm it was also important to include both positive and negative examples of the gesture. The reason for this was to ensure the algorithm also learned which movements did not belong to the gesture that was being trained. Thus, the research team had to ensure that gestures that the algorithm might confuse because they might look similar to the one that was currently being trained were included in the training set.

Algorithm training was performed for each gesture. This training process was identical for each of the 20 gestures listed in Table 1. The training used the video clips previously recorded with each actor. At the end of each training session, the VGB software by Microsoft returned the precision of the dataset. The resulting dataset precision ranges from 92.23–100%, with all gestures falling within this range. Table 2 displays algorithm precision by the gesture.

To test a gesture, video clips that were not used during training were to check gesture accuracy. The confidence value displayed by VGB for each of the video clips that were checked gives a value of 1, which means the gesture is recognized correctly.

Creating gestures is an iterative process involving doing several times to ensure high precision. The workflow diagram in Figure 1 below provides an overview of the build process used to create the dataset: In build step 1, gestures are recorded using Kinect Studio. In step 2 gestures are tagged using VGB. In step 3 the gestures are built and then analyzed in VGB. In step 4, the gestures are previewed in VgbView. The final step involves obtaining the database file and using it in the application.

For this study, the research team had to: (a) identify whether the gesture is correctly recognized by the Kinect sensor (successful gesture); and (b) the time taken to execute the gesture, which is measured from the moment the user is shown the gesture to the moment the gesture is recognized by the Kinect sensor. Both data are dependent on the physical and cognitive limitations of the user.

To obtain the data for execution times, the researchers developed a tool that measures the time taken to execute each gesture. The stop clock starts from the moment the icon of a gesture is displayed onscreen and ends when the Microsoft Kinect sensor recognizes the gesture.

## 5. Materials and Method

### 5.1. Description of Tool

The application DS_Sequences was used in the experiment run with participants to provide onscreen prompts and gather data for gesture *success rates* and *execution times*. The application DS_Sequences was used in the experiment run with participants to provide onscreen prompts and gather data for gesture success rates and execution times. The app was developed using Unity 3D v5.5.0f3 and C++ from Visual Studio 2015. The plugins used for app development were Kinect Unity add-ins and Kinect Visual Gesture Builder. Microsoft Kinect Visual Gesture Builder (VGB) has run-time gesture detection features that use the gesture database generated by VGB. This package will add settings that use the features of the native C++ project. All recognition was performed by the software, researchers only programmed gesture sequences and set the time limits established for the execution of each gesture.

The app allows researchers to configure basic routines for participants by selecting gestures from the catalog of 20 full-body gestures described in Table 1. It is important to note:Gestures can be repeated more than once in the same routine.Routines can be made as long or as short as needed by adding or removing gestures (see Figure 2a).Researchers may also decide how much time should be allocated to performing any given gesture (see Figure 2a).Times are logged for each gesture forming part of the routine.

In the case of this experiment, the research team selected 6 gestures for each routine and participants were given 10 s to complete each gesture.

Upon commencing a routine, the app delivers onscreen prompts to participants in the form of a gesture icon located in the top right of their screen (see Figure 2b). When the Kinect sensor identifies the gesture as having been executed correctly, the app DS_Sequences records the time taken to perform the gesture. In other words, it records the time taken from the moment the icon is displayed onscreen to the moment the sensor detects that the gesture has been executed correctly. In addition, the app also records the number of times gestures are performed successfully or not. A failed gesture is one that takes longer than 10 s to execute. The time available is displayed to participants on the top left of their screen (see Figure 2b).

In the app, the main screen is used to register new users, access the list of registered users, and consult the list of tasks that have been completed (see Figure 3a–c).

When creating new routines, researchers must allocate a name (test 01, test 02, etc.) to the new routine and indicate the time allowed (in seconds) for executing each gesture (Figure 2a). Researchers may create as many routines as they wish, ensuring each has the same number of gestures and the same time limits. Users can then be assigned any routine the research team deems necessary.

Before commencing the experiment and any sessions with participants, they were first registered as users in the app and their personal details were logged. Following this, each user was assigned their routines. Once the routines had been completed, the recorded data was exported in a CVS file for statistical processing in software packages. Figure 4 below shows the Entity Relationship Diagram (ERD) that displays data from the database and the relationships of entity sets.

### 5.2. Participants

The experiment was run using a sample size of 36 study participants with DS aged between 4 and 34 years old (mean value M = 18.78, standard deviation = 8.03). Fewer women participated in the study than men, totaling 31.56% of the sample. In total, 25 men and 11 women participated. The average age and standard deviation of female participants (M = 20.54, Sd = 6.20) was slightly higher than male participants (M = 18, Sd = 8.71). Participants were selected at random from two Down syndrome support centers and all participants had Trisomy 21 Down syndrome.

### 5.3. Methodology

#### 5.3.1. Equipment

The hardware used for the experiment consisted of a Kinect Sensor V2 and Kinect adaptor for Windows connected to a laptop via USB 3.0. The software application used to measure the time of gestures execution (DS_Sequences) was developed as previously mentioned in Section 5.1. During tests, the game was projected onto a larger screen connected to the laptop. Sessions were recorded using Camtasia Studio 8 (see Figure 5). All logs were stored in SQLite; data could then be exported from the database as a CVS file for subsequent analysis, as mentioned previously.

#### 5.3.2. Study Description

The study was performed in two Down syndrome support centers, both of which are located in the city of Monterrey in Mexico: ‘Centro Integral Down’ and the center ‘DIF de Santa Catarina’. To avoid any potential bias, participants were selected by the centers themselves based on the availability of those volunteering for the study. Before commencing the study, the research team informed all participants, family members, and the centers’ directors of the nature and objectives of the research in question to obtain informed consent. The study was conducted following the protocol approved by the Ethics Committee of the University of Monterrey.

Sessions were conducted for 4 h a day over 4 days at each center. Experimental tests were run with 8–10 participants per day. The psychotherapists of the respective centers supervised all sessions at all times.

Participants were verbally instructed that they would be asked to perform a short dance routine that involves following onscreen instructions in the form of a human silhouette. The research team then created gesture routines resembling dance choreographies within the application DS_Sequences, as shown in Figure 2a.

Although gestures were displayed onscreen in the form of an icon, a moderator also stood next to the participants to model the gesture (see Figure 6b) until they became familiar with the interface.

As shown in Figure 6a,b, users are presented with two pieces of information onscreen that provides them with real-time feedback as they attempt to perform gestures: firstly, a yellow silhouette on the bottom right of the screen that tracks their actions/gesture; and secondly, a countdown timer in the top left of the screen showing how much time they have left to complete the gesture.

The application records each of the routines, the number of completed gestures, and the time taken to complete them. Participants cannot continue with the choreography unless they correctly perform the gesture or the time runs out. The majority of participants were not able to complete the gestures within the assigned timeframe of ten seconds. (Table 3: 328 successful gestures vs. 425 failed gestures).

Once the data was collected, the CVS files were created and imported into the SPSS statistical program to perform the analysis required for the proposed research hypotheses.

## 6. Results and Analysis

To answer the research hypotheses of this study the research team analyzed the recorded data for the full-body gestures executed by study participants. At this point, it is important to remember that a gesture executed in less than 10 s is considered successful, whereas one that exceeds 10 s—either because it was executed incorrectly or because the user’s physique is not a perfect fit with the pattern the Kinect sensor is looking for—is considered a failed gesture.

Firstly, gestures were compared based on rates of success and failure, and then the variable gender was analyzed to establish how it might affect gesture success rates. Next, the execution times for each gesture were analyzed to identify whether there was a significant difference between the gestures; in other words, to establish whether some gestures can be executed faster than others. Finally, the research team analyzed whether the variable gender affects the time taken to execute a task; in other words, whether male participants or female participants perform gestures at the same speed, or whether one gender executes gestures faster than the other.

### 6.1. Comparison of Gestures by Success Rate

Drawing from the data presented in Figure 7, it is possible to assert that the following gestures were executed with a greater degree of success: (#14) Raise both arms and bend elbows, (#10) *Raise right arm high over head*, (#11) *Raise left arm high over head and* (#5) *Crucifixion*; *whilst the gestures that were executed to a lesser degree of success were* (#4) *‘A’ shape. Raise arms slightly to form an ‘A’*, (#15) *Raise both hands to head*, (#16) *Place hands on hips*, *and* (#20) *Bend forward*. Upon reviewing these same gestures by Gender (see Figure 8), it was observed that male participants executed gestures #17 and #20 successfully more often than female participants. The statistical analysis was performed using Pearson’s Chi-Square (χ^2^). The following step consisted of comparing all twenty gestures according to the number of successful gestures and the number of failed gestures, as well as success rates by Gender.

As we are dealing with a dichotomous variable (success or failure) and not following a normal distribution, the nonparametric Kruskal–Wallis test was applied. Table 3 shows the percentages of success rates and failures for each gesture. The χ^2^ test measures the discrepancies between two measurements. A value of χ^2^ = 56.560 was obtained with *p*-value = 0.000, which indicates that there is a significant difference between success and failure in every gesture there are significant differences between the success rates and failure rates for each gesture (see Table 4). Upon observing the contingency table (Table 3) it can be argued that the gestures that have a higher percentage of success are significantly better (marked with *). Based on these results the null hypothesis H_01_ is rejected, therefore, *H1 is accepted:* The presence or absence of Kinect recognition logs indicates that the different full-body gestures being executed by individuals with DS present different levels of complexity.

### 6.2. Comparison of Gestures by Gender

The next step is to verify whether a person’s gender has a direct impact on gesture execution. Figure 8 shows the success rate of Kinect gestures by Gender. Table 5 displays the global results for successful and failed gestures by Gender.

In Table 6, the analysis using Pearson’s Chi-Square (χ^2^) indicates that when looking at overall data by *Gender*, there is a significant difference between the success rates and failure rates for gesture execution, χ^2^ = 16.94 *p*-value = 0.000. According to the data in Table 5, the success rate is higher for male participants, but for both males and females there are a greater number of overall fails compared to successes. This indicates that individuals with DS experience considerable difficulty executing full-body gestures. The null hypothesis H_02_ is rejected, therefore, *H2 is*
*accepted*: The execution success of full-body gestures is influenced by gender. It is believed that there is a significant difference between the rates of success and failure for gesture execution in terms of Gender.

In observing only the values for success and failure of each gesture, it is possible to observe greater success in gestures executed by male participants than female participants. To demonstrate H3 and H4 the calculation is focused on the success rates for each gesture, and the Kruskal–Wallis test is applied for the variables *Success* and *Gender*.

The results obtained for success were χ^2^ = 8.611 and *p*-value = 0.979, whilst for gender the results obtained were χ^2^ = 0.000 and *p*-value = 0.999. What this indicates is that there is no significant difference for either variable. That is to say, all gestures have the same degree of success and no gesture stands out, nor is there a significant difference in terms of gender. As such, the null hypothesis H_03_ and H_04_ are accepted. Generally speaking, it is possible to state that the success of gesture execution does not depend on the gender of users.

### 6.3. Comparison of Gesture Success Rates by Execution Times

Lastly, execution times were analyzed to determine whether there were differences by gesture. Figure 9 below shows the findings for each gesture. As can be observed, the gestures that took the longest to execute were (#20) *Bend forward*, (#4) *‘A’ shape. Raise arms slightly to form an ‘A’*, and (#8) *Raise arms in front of body*, whereas gestures (#10) *Raise right arm high over head* and (#11) *Raise left arm high overhead* were recognized the fastest.

Timings follow a normal distribution; as such, a parametric test has to be applied. All ANOVA comparisons indicate that there is no significant difference in each group (all *p*-values are greater than 0.05). Levene’s test for equality of variance produces *p*-value = 0.005, thus making it necessary to perform the nonparametric Kruskal–Wallis Test.

The nonparametric Kruskal–Wallis test (see Table 7) indicates that there is no significant difference between the execution times of each gesture χ^2^ = 26.488 *p*-value = 0.117. As the time taken to execute the different gestures is the same statistically speaking, the null hypothesis H_05_: All gestures take the same time to execute is accepted, therefore, research Hypothesis *H5 is rejected,* and it is possible to state that all successful gestures are executed equally fast.

In considering only the execution times for each gesture by gender and applying the Kruskal–Wallis test, the following results were obtained: χ^2^ = 0.000 and *p*-value = 0.789. What this indicates is that there is no significant difference in executions times by gender. That is to say, all gestures take the same length of time to execute regardless of a participant’s gender. Based on this finding, the null hypothesis H_06_: Gender does not influence the execution time of full-body gestures, is accepted. Consequently, research hypothesis *H6 is rejected*, thus affirming that the time taken to execute a gesture does not depend on the user’s gender.

## 7. Conclusions

Further to the data and results that have been presented, observational findings were made during the experiment that requires discussion. It was found that the verbal instructions given to participants before commencing sessions were insufficient. Consequently, the moderator had to physically demonstrate how to perform the gestures so that the study participants understood how they were expected to execute gestures during their routines.

Neurotypical individuals and individuals with DS execute gestures differently. Generally speaking, gestures designed for the Kinect sensor are developed using neurotypical individuals. As such, when individuals with physical or cognitive impairments have to input instructions using gestures, yet execute the gestures differently, systems do not respond as they should. Logically, this causes frustration amongst such users and a lack of trust in technological systems.

Gesture databases can be created ad-hoc, taking into account specific considerations. But in general terms, and from a Design for All (DfA) perspective, the gestures used in the recognition databases must be inclusive; this is to say, the sensor must be able to recognize them regardless of who is performing them.

For this study, a database containing 20 gestures was developed using neurotypical actors. During the fieldwork, individuals with DS executed the proposed gestures. Using the Kinect sensor, the researcher team was able to establish differences in gesture execution.

During the fieldwork, the observations made alerted the researchers to two issues: firstly, the Microsoft Kinect did not detect certain gestures, even when they seemed to be correctly executed by participants. This was particularly evident with gesture (#4): *‘A’ shape. Raise arms slightly to form an ‘A’*. Participants’ arms would frequently fall at a slightly more acute or obtuse angle than required, perhaps as a result of cognitive deficits in spatial ability, or a lack of muscle mass that would allow the participants to hold their arms at the requested angle. This lead to gestures being performed in a different pattern to the pattern created by the neurotypical person which led to them not being recognized by the sensor; secondly, it was observed that people with DS similarly perform movements but different to those performed by a neurotypical person. In other words, just as individuals with DS have a characteristic gait, the same is true for some gestures. The way gestures are executed is influenced by the physical impairments they share in common. For example, it is possible that the curvature of the spine, which is characteristic of people with DS, affects the way they lift their arm as they attempt to raise it to 90°. Based on this, it is possible to state that users with Down syndrome are not able to correctly execute some gestures designed by and for neurotypical individuals, and the Microsoft Kinect therefore cannot recognize them.

Taking into consideration the execution of all gestures studied in this work, it is possible to state that the number of failed gestures predominates over the number of successful recognition gestures. Gender does not influence the success or failure of a gesture; both male participants and female participants produced more failed gestures than successful gestures. As such, it is possible to confirm that both men and women execute gestures in the same manner. However, in focusing only on the number of successful gestures and comparing the type of gesture, it is possible to state that all gestures have the same success rate, no gesture stands out and there is no significant difference in terms of gender; in other words, both men and women execute gestures with the same degree of success.

With regards to time taken to execute gestures, it is possible to state that for the study population all gestures took the same length of time to execute regardless of gender.

What this indicates is that individuals living with DS perform gestures differently. The Microsoft Kinect sensor highlights this fact when gestures performed by individuals with DS are compared against the gesture patterns created by neurotypical individuals.

Despite the obvious differences, the results generated by the study revealed that certain gestures performed by individuals with DS were recognized by the Microsoft Kinect (database of gestures created), and of these, some were recognized faster than others; in other words, some gestures are easier for individuals with DS to perform than others. We understand this to be the consequence of the physical and cognitive impairments that these individuals display.

The gestures with higher success rates were (#14) Raise both arms and bend elbows, (#10) Raise right arm high over head, (#11) Raise left arm high over head, and (#5) Crucifixion: Raise both arms out to the side at a 90° angle. The gesture that was the most difficult to recognize was (#4) A shape: Raise arms slightly to form an ‘A’. As mentioned before, although it seemed as though participants executed this gesture correctly, how the arms were placed made it impossible for the Kinect sensor to recognize the gesture and it was therefore recorded as a failed gesture. Although the execution of different gesture and gender do not significantly influence execution times, it is possible to state that the fastest gestures were (#18) Raise right arm to head and (#19) Raise left arm to head, whilst those that were the slowest to execute/recognize were (#20) Bend forward, (#4) A shape: Raise arms slightly to form an ‘A’, and (#8) Raise arms in front of body.

Numerous studies have tested the versatility and utility of Microsoft’s Kinect sensor. These studies have shown how it can be used for therapeutic purposes and provide therapeutic benefits to individuals with different illnesses or disabilities. However, when we are talking about gestures, it is important to remember that the Kinect sensor ties to identify a gesture by comparing what is detected against one that has previously been built as a reference model. For this reason, individuals with physical and/or cognitive impairments may find videogames and applications that gather input data from gestures frustrating, as how they perform the gestures can differ slightly to the model gesture sets that are normally created by neurotypical individuals. This is an important fact to keep in mind when developing applications or videogames in which the Kinect sensor is used as the input device for HCI.

The contribution made by this study is to identify which gestures the Kinect can recognize, regardless of who performs them (neurotypical individuals or individuals with DS); in other words, the authors establish which gestures, from those analyzed, are inclusive. In line with this, the authors also argue that it is vital for application programmers to set up inclusive gesture databases, which is achieved by selecting gestures that can be recognized by the sensor, no matter which type of user may be executing them.

Nonetheless, a guide that provides an assessment of the gestures that were the object of this study has been included in Appendix A of this paper. The target audience of the guide is software programmers and developers. The aim is to provide these professionals with a tool that will allow them to select the most suitable gestures of their app, thereby ensuring the app is also inclusive and accessible for people with DS. The guide includes the following details:Name: Nomenclature used to refer to a gesture. General gestures are known by a specific name. If a name does not exist, describe it as clearly as possible so it is possible to visualize it without looking at its image.Objective: All gestures have a function, some more than one.Gesture: Image of the gesture so it is easy to identify visually.Ease of use: Rating using one to four stars; 4-stars indicates the gesture is very easy to perform. It is an easy and fast way to visually identify whether the gesture is recommendable or not.Description of problem: A short description that explains what problems could arise. This field can help the programmer use the gesture whilst avoiding difficulties.Recommendation: A summary of how and when to use the gesture.

## Figures and Tables

**Figure 1 sensors-20-03930-f001:**
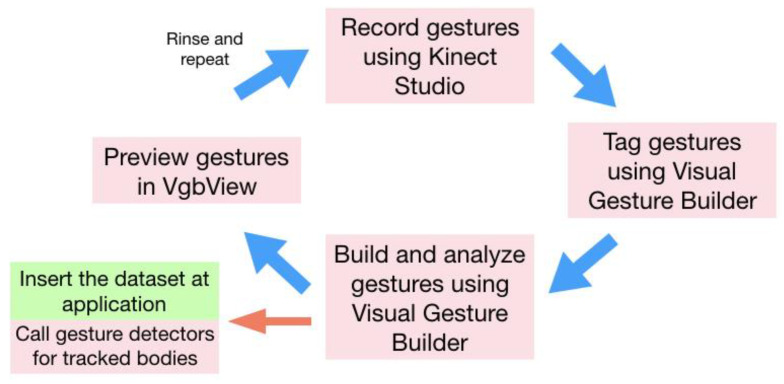
Overview of the build process used to create a dataset [43].

**Figure 2 sensors-20-03930-f002:**
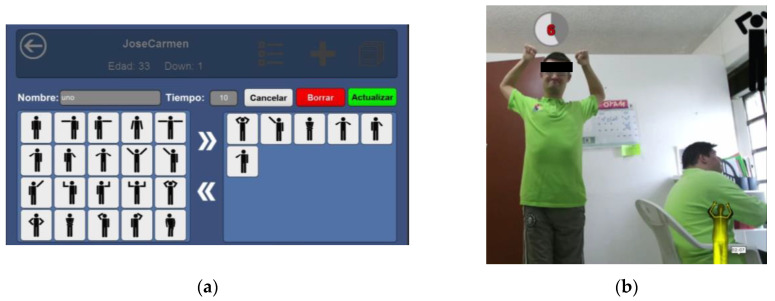
(**a**) Designing a Kinect routine; (**b**) Participant performing gesture 15 ‘Raise both hands to head’.

**Figure 3 sensors-20-03930-f003:**
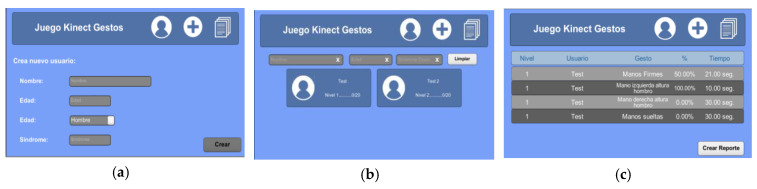
(**a**) User registration; (**b**) list of registered users; (**c**) list of tasks performed.

**Figure 4 sensors-20-03930-f004:**
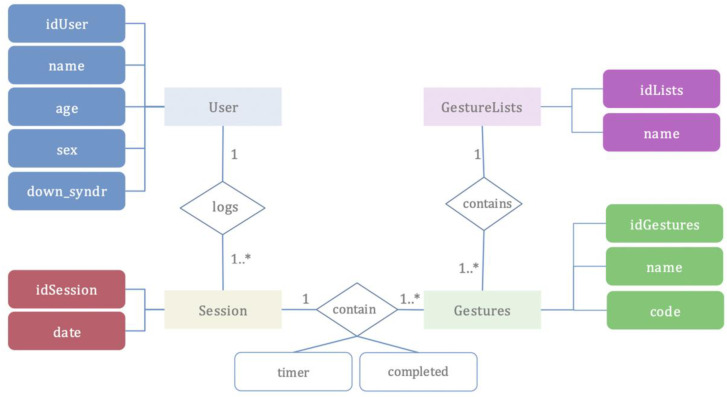
Entity Relationship Diagram (ERD): Database.

**Figure 5 sensors-20-03930-f005:**
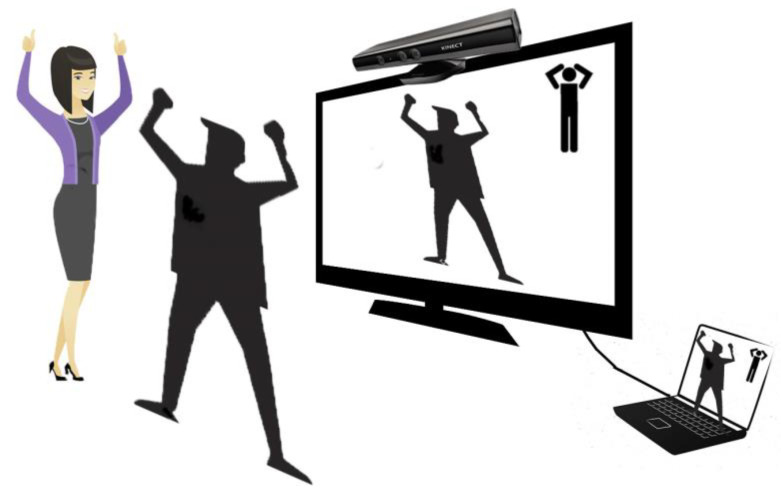
Set up for gesture testing sessions.

**Figure 6 sensors-20-03930-f006:**
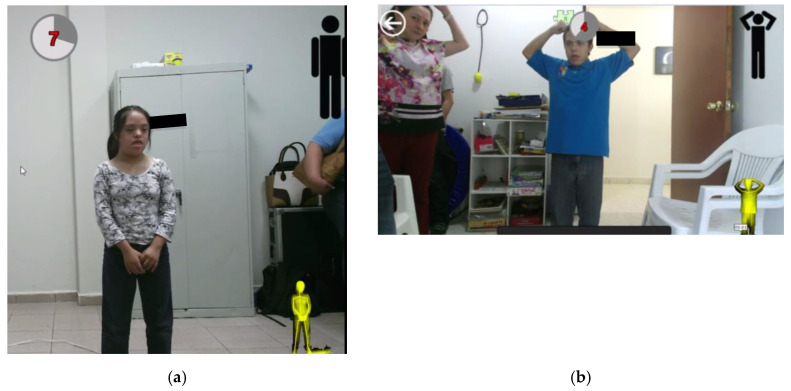
(**a**) Participant attempting to perform a gesture (1); (**b**) moderator demonstrating gesture to a participant.

**Figure 7 sensors-20-03930-f007:**
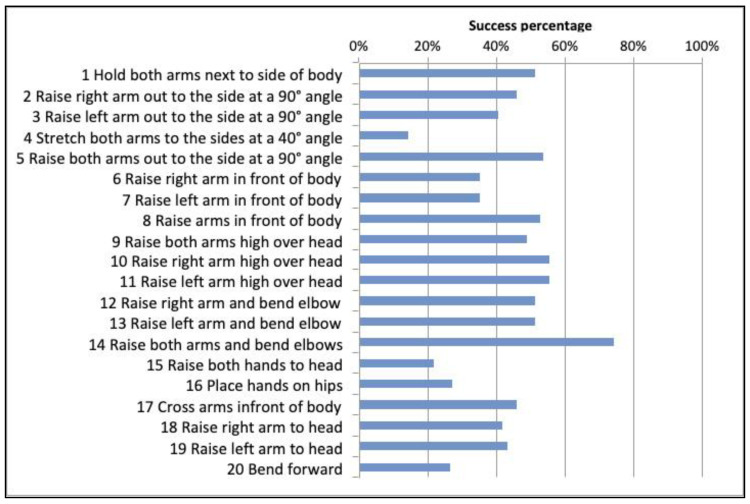
Success rate percentages by Kinect gesture.

**Figure 8 sensors-20-03930-f008:**
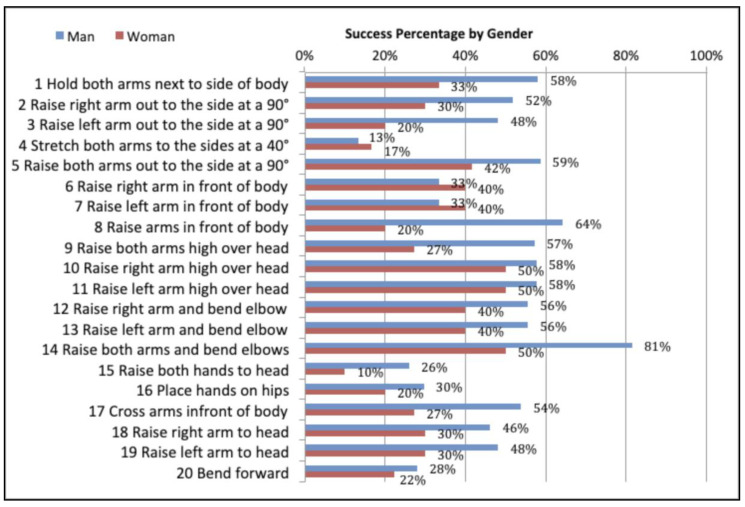
Success rate of Kinect gestures by gender.

**Figure 9 sensors-20-03930-f009:**
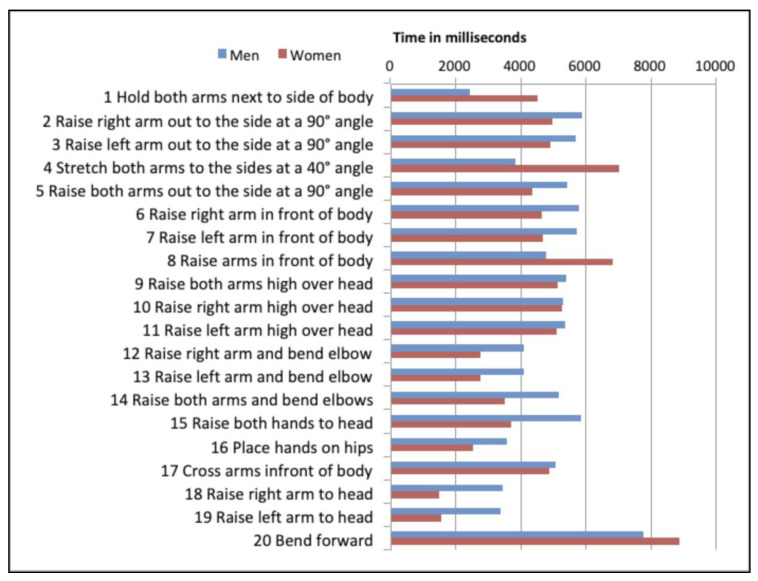
Gestures by execution time and gender.

**Table 1 sensors-20-03930-t001:** Physical gestures selected to study the User Interface (UI).

Gesture Name	Icon	Visual Description	Gesture Name	Icon	Visual Description
(#1) Hold both arms next to side of body	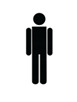	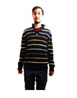	(#11) Raise left arm high over head	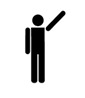	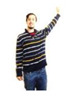
(#2) Raise right arm out to the side at a 90° angle	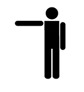	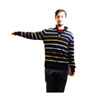	(#12) Raise right arm and bend elbow	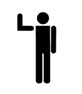	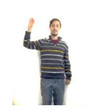
(#3) Raise left arm out to the side at a 90° angle	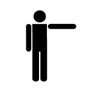	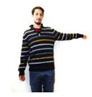	(#13) Raise left arm and bend elbow	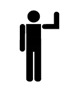	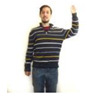
(#4) A shape. Raise arms slightly to form an ‘A’.	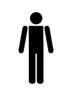	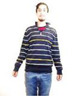	(#14) Raise both arms and bend elbows	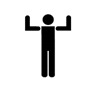	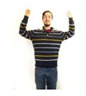
(#5) Crucifixion. Raise both arms out to the side at a 90° angle	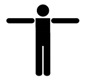	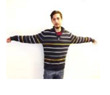	(#15) Raise both hands to head	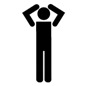	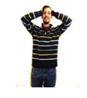
(#6) Raise right arm in front of body	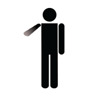	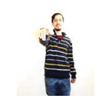	(#16) Place hands on hips	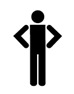	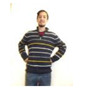
(#7) Raise left arm in front of body	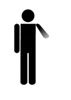	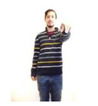	(#17) Cross arms in front of body	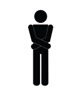	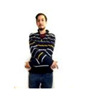
(#8) Raise arms in front of body	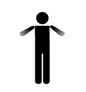	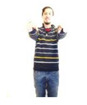	(#18) Raise right arm to head	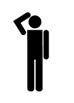	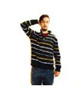
(#9) Raise both arms high over head	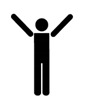	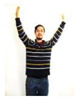	(#19) Raise left arm to head	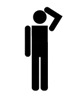	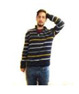
(#10) Raise right arm high over head	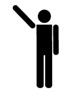	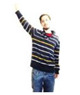	(#20) Bend forward	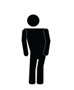	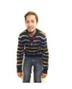

**Table 2 sensors-20-03930-t002:** Recognition rate according to the gesture type.

Gesture Name	Accuracy (%)	Error (%)	Gesture Name	Accuracy (%)	Error (%)
Gesture (#1)	100.00	0.00	Gesture (#11)	97.78	0.34
Gesture (#2)	98.10	1.10	Gesture (#12)	95.33	3.33
Gesture (#3)	97.69	1.56	Gesture (#13)	95.50	2.83
Gesture (#4)	96.58	2.23	Gesture (#14)	96.88	1.34
Gesture (#5)	100.00	0.00	Gesture (#15)	94.45	2.32
Gesture (#6)	94.89	3.45	Gesture (#16)	93.99	3.43
Gesture (#7)	94.50	2.87	Gesture (#17)	93.93	5.50
Gesture (#8)	93.12	3.34	Gesture (#18)	95.23	1.92
Gesture (#9)	98.24	0.24	Gesture (#19)	94.84	2.23
Gesture (#10)	96.45	0.31	Gesture (#20)	92.23	3.44

**Table 3 sensors-20-03930-t003:** Contingency tables for success rates of Kinect gestures.

Gesture	No. Successful Gestures	No. Failed Gestures	No. Total Gestures	Percentage of Successful Gestures	Percentage of Failed Gestures
(#1) Hold both arms next to side of body	22	21	43	51.2% *	48.8%
(#2) Raise right arm out to the side at a 90° angle	17	20	37	45.9%	54.1%
(#3) Raise left arm out to the side at a 90° angle	15	22	37	40.5%	59.5%
(#4) A shape. Raise arms slightly to form an ‘A’.	6	36	42	14.3%	85.7%
(#5) Crucifixion. Raise both arms out to the side at a 90° angle	22	19	41	53.7% *	46.3%
(#6) Raise right arm in front of body	13	24	37	35.1%	64.9%
(#7) Raise left arm in front of body	13	24	37	35.1%	64.9%
(#8) Raise arms in front of body	20	18	38	52.6% *	47.4%
(#9) Raise both arms high over head	20	20	40	50.0% *	50.0%
(#10) Raise right arm high over head	20	16	36	55.6% *	44.4%
(#11) Raise left arm high over head	20	16	36	55.6% *	44.4%
(#12) Raise right arm and bend elbow	19	18	37	51.4% *	48.6%
(#13) Raise left arm and bend elbow	19	18	37	51.4% *	48.6%
(#14) Raise both arms and bend elbows	26	9	35	74.3% *	25.7%
(#15) Raise both hands to head	8	29	37	21.6%	78.4%
(#16) Place hands on hips	10	27	37	27.0%	73.0%
(#17) Cross arms in front of body	17	21	38	44.7%	55.3%
(#18) Raise right arm to head	16	21	37	43.2%	56.8%
(#19) Raise left arm to head	16	21	37	43.2%	56.8%
(#20) Bend forward	9	25	34	26.5%	73.5%
**Total**	**328**	**425**	**753**	**43.6%**	**56.4%**

**Table 4 sensors-20-03930-t004:** Pearson’s Chi-square (χ^2^) Test for Kinect gestures.

	Value	gl	Asymp. Sig. (2-Sided)
Pearson’s Chi-Square	56.560 *	19	0.000
Likelihood Ratio	59.755	19	0.000
Linear-by-Linear Association	0.206	1	0.650
No. Valid Cases	753		

* 0 cells (0.0%) have expected count less than 5. The minimum expected count is 14.81.

**Table 5 sensors-20-03930-t005:** Contingency table.

Gender	No. Successful Gestures	No. Failed Gestures	No. Total Gestures	Percentage of Successful Gestures	Percentage of Failed Gestures
Men	263	285	548	48.0%	52.0%
Women	65	140	205	31.7%	63.3%
Total	328	425	753	43.6%	56.4%

**Table 6 sensors-20-03930-t006:** Pearson’s Chi-square (χ^2^) Test for differences by gender.

	Value	gl	Asymp. Sig. (2-Sided)	Exact Sig. (2-Sided)	Exact Sig. (1-Sided)
Pearson’s Chi-Square	16.094 **	1	0.000		
Continuity Correction *	15.438	1	0.000		
Likelihood Ratio	16.440	1	0.000		
Fisher’s Exact Test				0.000	0.000
No. Valid Cases	753				

* Computed only for a 2 × 2 table. ** 0 cells (0.0%) have expected count less than 5. The minimum expected count is 89.30.

**Table 7 sensors-20-03930-t007:** Contrast statistics (Kruskal–Wallis test, grouping variable: ID_gesture).

	Execution Times for Successful Gestures
Pearson Chi-Square	26.488
gl	19
Asymp. Sig.	0.117

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
