# Peer review of "Evaluation of Full-Body Gestures Performed by Individuals with Down Syndrome: Proposal for Designing User Interfaces for All Based on Kinect Sensor"

_sensors, 2020, doi:10.3390/s20143930_

Round 1
Reviewer 1 Report
The paper main conclussion is that kinect sensor cannot be applied to people with DS.
My main argue is that the approximation is not correct. The approximation should be I have detected a problem with kinect sensor with people with some disabilities, so i include a new gesture database for solving this problem. So researchers can use my database for this can of use. The second part is not addressed.
It include some pictures with faces. They should be blurred.
Author Response
We thank the reviewer for the valuable comments and suggestions that help us improve the quality of our paper. We have carefully revised the paper reflecting on them. We provide a file attached with our responses to the reviewer’s questions and comments.

Reviewer 2 Report
I believe that the current version of the article addressed satisfactorily all the issues identified by the reviewers.
The overall clarity of the presentation has been significantly improved.
In my opinion, the article can be accepted for publication as it is now.
Author Response
Thank you for your selfless work in reviewing this manuscript.
Your comments throughout the review process have proven invaluable in helping to disseminate this research in this paper.

Reviewer 3 Report
My comments and questions are given in the attached file.

Author Response
We thank the reviewer for the valuable comments and suggestions that help us improve the quality of our paper. We have carefully revised the paper reflecting on them. We provide a file attached to our responses to the reviewer’s questions and comments.

Round 2
Reviewer 1 Report
The authous have addressed my suggestions
This manuscript is a resubmission of an earlier submission. The following is a list of the peer review reports and author responses from that submission.
Round 1
Reviewer 1 Report
The most annoying issue of this report is the permanent confusion between "gesture execution" and "gesture recognition". Therefore the significance of the terms "successful gesture" and "failed gesture" is not clear. Considering the fact that the executants of the gestures are people with serious cognitive impairments, before thinking of gesture recognition it is important to determine whether the respective gestures were correctly executed.
This is important for defining the moments when the execution and recognition phases begin and end. The authors state that "The stop clock starts from the moment the icon of a gesture is displayed on screen and ends when the Microsoft Kinect sensor recognizes the gesture." It seems that the software application used in the experiment measures only the duration between the moment when the user is prompted to execute a gesture (by displaying the icon on the screen), and the moment when the Kinect software reports the end of the recognition phase. In other words, the measured duration is the duration of the whole process comprising both the execution and recognition phases.
This makes the entire experiment questionable because the execution phase takes several seconds, while the recognition phase takes only a few milliseconds. Therefore the experimental data does not reflect the capability of the Kinect system to recognize gestures, but the ability of the people involved in the experiment to correctly execute the required gestures.
I also note that the title does no really reflect the content of the paper (it should contain the keyword Dawn Syndrome)
The English presentation needs a careful review. Even the first phrase in the abstract (lines 11-13) is confusing.
Author Response
We thank the reviewer for the valuable comments and suggestions that can help us improve the quality of our paper. We have carefully revised the paper reflecting them. We provide our responses to the reviewer’s questions and comments in the attached file.

Reviewer 2 Report
The paper deals with full body gesture recognition for users with Down syndrome.
The main conclussion is that kinect sensor is not appropiate due to their physical imapirements.
The methodoly regarding algorithms or maths for the patter recognition is missing.
I recommend major revision
Author Response

(The authors gave the same response as above.)

Reviewer 3 Report
My comments and questions are given in the attached file.

Author Response

(The authors gave the same response as above.)

Round 2
Reviewer 1 Report
The authors properly addressed all the issues identified in my previous review.
I believe that the current version of the report may be considered for publication.
Author Response
Reviewer #1
Thank you for your final review of the paper, which you now consider apt for publication.
Your valuable comments and suggestions helped to improve the manuscript.
Reviewer 2 Report
The paper deals with full body gesture recognition for users with Down syndrome.
The main conclussion is that kinect sensor is not appropiate due to their physical imapirements so the system takes more time to detect
The methodoly regarding algorithms or maths for the patter recognition is still missing but my main concern is about the orientation of the paper. Instead of pointing the not goodness of the sensor I would orientate it to reprogramming the kinect libraries for the correct detection in DS, or identifying another sensor for this group of people.
I recommend rejection with present orientation.
Author Response
Reviewer #2
Thank you for your last review of our work. Your valuable comments and suggestions have helped to improve the manuscript.
Your comments have been taken into account and addressed in the conclusions section. We are sorry that the focus of the article (UI / UX study), is not entirely to your satisfaction, but, as we mentioned, we feel that too much “noise” would be introduced by adding the mathematical foundations and algorithms, and thus distract from the actual focus of our work. We would rather focus our attention on gesture programming in a separate article.
Reviewer 3 Report
Thank you for addressing my comments and questions from the previous iteration. After the corrections made to the current version, the authors have resolved all my concerns and improved their paper.
Author Response
Reviewer #3
Thank you for your work in reviewing this paper. We would like to sincerely thank you for the effort that was put into it.
Your valuable comments and suggestions helped to improve the manuscript.